# Predictive Optimization of Electrical Conductivity of Polycarbonate Composites at Different Concentrations of Carbon Nanotubes: A Valorization of Conductive Nanocomposite Theoretical Models

**DOI:** 10.3390/ma14071687

**Published:** 2021-03-30

**Authors:** Lakhdar Sidi Salah, Nassira Ouslimani, Mohamed Chouai, Yann Danlée, Isabelle Huynen, Hammouche Aksas

**Affiliations:** 1Research Unit Materials, Processes and Environment (URMPE), Faculty of Technology, M’Hamed Bougara University, Boumerdes 35000, Algeria; h.aksas@univ-boumerdes.dz; 2Processing and Shaping of Fibrous Polymers Laboratory, Faculty of Technology, University M’Hamed Bougara of Boumerdes, Avenue of Independence, Boumerdes 35000, Algeria; n.ouslimani@univ-boumerdes.dz; 3Signals and Systems Laboratory, Department of Electrical Engineering, Mostaganem University, Site 1 Route Belahcel, Mostaganem 27000, Algeria; mohamed.chouai@univ-mosta.dz; 4ICTEAM Institute, Université Catholique de Louvain, 1348 Louvain-la-Neuve, Belgium; yann.danlee@uclouvain.be (Y.D.); isabelle.huynen@uclouvain.be (I.H.)

**Keywords:** polycarbonate (PC), carbon nanotubes (CNTs), electrical conductivity, hyperparameter, optimization, percolation, microwave

## Abstract

Polycarbonate—carbon nanotube (PC-CNT) conductive composites containing CNT concentration covering 0.25–4.5 wt.% were prepared by melt blending extrusion. The alternating current (AC) conductivity of the composites has been investigated. The percolation threshold of the PC-CNT composites was theoretically determined using the classical theory of percolation followed by numerical analysis, quantifying the conductivity of PC-CNT at the critical volume CNT concentration. Different theoretical models like Bueche, McCullough and Mamunya have been applied to predict the AC conductivity of the composites using a hyperparameter optimization method. Through multiple series of the hyperparameter optimization process, it was found that McCullough and Mamunya theoretical models for electrical conductivity fit remarkably with our experimental results; the degree of chain branching and the aspect ratio are estimated to be 0.91 and 167 according to these models. The development of a new model based on a modified Sohi model is in good agreement with our data, with a coefficient of determination $R^{2}=0.922$ for an optimized design model. The conductivity is correlated to the electromagnetic absorption (EM) index showing a fine fit with Steffen–Boltzmann (SB) model, indicating the ultimate CNTs volume concentration for microwave absorption at the studied frequency range.

## 1. Introduction

Recently, carbon nanotube-based composites have gained large interest as conductive fillers used in the synthesis of a wide scale of conductive polymer composites (CPCs) by extrusion process; these CPCs are used in many fields such as anti-static materials for electrostatic discharge (ESD) shielding, electromagnetic interference (EMI) shielding, sensor and conductors [1,2]. The electrical conductivity of carbon nanotubes (CNTs) is tremendously higher than that of insulating polymers; CNTs have a large aspect ratio and complicated microstructural, and physical issues regarding compounded blends make the prediction of the effective conductivity of polymer-CNT nanocomposites a difficult task [3]. The prediction and modeling of the electrical behavior of composite materials are considered advantageous when an optimized design can be adapted to any particular application instead of using multiple cost and onerous experimental studies [4]. Several theoretical models have been applied to predict the conductivity of the polymer–nanofiller composite systems [5,6]. Their validity and limitations are checked, and on the other hand, the development of a new model for the prediction of DC conductivity has also been investigated by Rahman et al., who strongly reinforced the interpretation of the results based on the transitional percolation threshold [7]. Several studies reported the use of direct current (DC) electrical conductivity as a powerful framework for predicting the physical properties of materials embedded with nanoparticles [8,9]. Jinkai Yuan et al. [10] have initially applied power percolation theory using alternating current (AC) electrical conductivity as a way of predicting the critical weight content of interconnected nanoparticles responsible for the change of the physical properties of the elaborated heterogenized nanomaterial. This latter study gives a new area of modeling the percolation threshold over frequency, as well as upgrading the use of theoretical electrical conductivity as empirical models to predict the AC conductivity of composites, as investigated by Bouknaitir et al. [11]. Ranvijai Ram et al. [12] reported the dependency of electrical conductivity and EMI shielding effectiveness on the type, weight content and geometric characteristics of the filler as aspect and surface to volume ratios using different fundamental theoretical and mathematical models, such as the Steffen–Boltzmann (SB) model; this latter has been used for predicting efficiently the percolation threshold exploring electromagnetic properties, as investigated by Rahaman et al. [13]. One of the main challenges regarding EMI shielding relates to the design of a material with high conductivity and a low thickness, aiming at maximizing EMI, in order to make it compatible with microwave targeted components [14].

In our study, we investigate the use of theoretical models to predict and simulate the electrical conductivity of polycarbonate—carbon nanotube (PC-CNT) composites. The originality of our work is folded in: I. the use of hyperparameter optimization as a way to predict the average AC conductivity of PC-CNT composites measured over frequency range 15–25 GHz. II. the use of measured AC conductivity as a parametric tool to estimate the CNT nanofiller physical characteristics, such as the aspect ratio, surface to volume ratio of CNTs and the extent of conducting CNTs chains in polycarbonate (PC) thin films. III. checking the applicability of various theoretical models through their validation by comparison with experiments. The combination of conductivity models and electromagnetic absorption model aims to design an ultrathin wideband microwave absorber operating in the 15–25 GHz frequency range, using the optimum weight concentration of CNT nanofillers according to the absorption ratio curve.

## 2. Materials and Methods

### 2.1. Materials Used and Blending Process of PC-CNT Conductive Composites

The elaboration of our conductive composites required the use of polycarbonate Bayer Makrolon^®^ OD2015 as an insulating polymer matrix, and as conductive nanofillers multi-walled NC7000 carbon nanotubes (MWCNTs, from NanoCyl SA, Sambreville, Belgium) produced by catalytic chemical vapor deposition (CCVD) method; their physical properties are reported in [15]. The densities of PC and MWCNTs are 1.19 g/cm^3^ and 1.75 g/cm^3^ respectively. CNTs are added in an appropriate weight or equivalent volume concentration, as presented in Table 1. The composite compounds are melt-blended at 280 °C for 5 min at 150 RPM in a micro 15 DSM micro-compounder. Composite pellets are twice hot-pressed in Fontijne press under 290 °C, 7.5 T, and 2.5 min to produce films with 140 µm thickness.

### 2.2. Observation and Characterization

The morphology of microtome PC-CNT composites cut into 100 nm slices, using a diamond knife, was studied with the help of LEO-922 transmission electron microscope (TEM) at an accelerating voltage of 120 kV.

The respective electrical conductivity for different volume fractions of CNTs is obtained from electromagnetic characterization using Anritsu M54644B Vector Network Analyzer (VNA) in waveguide configuration [16]. Electromagnetic parameters as conductivity were characterized over the Ku and K bands with a focus on 15–25 GHz frequency range. The calibration was made by LRL/LRM [17] method for each frequency band, and the IF bandwidth was set at 300 Hz.

### 2.3. Hyperparameter Optimization

In this study, we have used hyperparameter optimization [18] using a grid-search strategy. An exhaustive search is performed on a manually specified subset of the hyperparameter space of the training algorithm. A grid search should be done through a performance metric, which is usually calculated by the mean square error (MSE) [19]. We have focused on the need to automatically tune and optimize those physical parameters in order to obtain the best combination that maximizes the tested model performance as assessed by the calculation of mean square error.

### 2.4. Mean Square Root Error and Coefficient of Determination Methods for Model Fitting

We have performed a checking of the quality of the fitting curves in order to assess the quality of the hyperparameter optimization process. It allows a good identification of the most appropriate model for predicting the electrical conductivity of PC charged by CNTs. Coefficient of determination (*R*^2^) is used to measure the precision of the fitting of our predicted plots with the experimental values; we have calculated *R*^2^ based on the ratio between the sum of squares of the regression (SSR) and the total sum of squares (SST) [20] given by the following equation:(1a)$$R^{2}=\frac{\mathrm{SSR}}{\mathrm{SST}}=\frac{\sum_{i=1}^{n}{\left(\bar{y_{i}}-\bar{y}\right)}^{2}}{\sum_{i=1}^{n}{\left(y_{i}-\bar{y}\right)}^{2}}$$
where $\bar{y}$ is the average of the data set; the quality of the fit is ascribed to the value of the mean square error (MSE); the lower the MSE, the better the system performance.

The equation of MSE we have used in modeling the exponent factor k of Mamunya model as shown in Section 3.2.4 is described as follow [21]:(1b)$$\mathrm{MSE}=\frac{1}{n}\sum_{i=1}^{n}{\left(y_{i}-\bar{y_{i}}\right)}^{2}$$
where n: number of point of data; y_i_ and $\bar{y_{i}}$ is the observed and predicted value of electrical conductivity obtained by experimental procedure and Mamunya model, respectively.

## 3. Results and Discussion

### 3.1. Theoretical Models Background and Applicability

#### 3.1.1. Determination of Percolation Using Power Law Theory

The conductive composites have been characterized by a VNA for a percolation analysis in microwave frequencies. Figure 1 depicts the AC conductivity values of PC-CNT films plotted as a function of volume fraction vol.% CNT loading. The reported values are the average conductivity over the frequency range 15–25 GHz. A three-stage transitional behavior is seen for PC-CNT composites initialed by a low increase in electrical conductivity, followed by a very sharp increase in conductivity and an upsurge of almost three orders in magnitude is evaluated at 1 vol.% CNTs, indicating the formation of a percolating network, i.e., a continuous path. Once the first conductive network is formed, the addition of extra fillers beyond percolation increases only the number of such conductive networks, but does not significantly affect the conductivity as depicted by the final stage [13]. The percolation threshold has been predicted by plotting the conductivity as a function of the CNT loading. The empirical data fit basically used in DC is applicable in microwave range, since the behavior is similar [15,22]:(2)$$\sigma \;=\;k{\left(\phi -\phi_{c}\right)}^{\mu }\cdot \mathrm{for}\cdot \phi >\phi_{c}$$
where k and μ are fitting constants, ϕ is the volume fraction of reinforcement and $\phi_{c}$ is the percolation threshold [22]. The equation can be written by taking the logarithm of both sides:(3)$$\log(\sigma )=\log\left(k\right)\;+\;\mu \log\left(\phi -\phi_{c}\right)$$

By taking the linear regression of the plot $\log\left(\sigma \right)$ vs. $\log\left(\phi -\phi_{c}\right)$, both the percolation threshold and critical exponent can be evaluated. As the percolation threshold is the minimum filler content where the first continuous network of filler particles is formed within the polymer matrix, both a lower percolation threshold and higher critical exponent are typically an indication of a homogenous dispersion of filler within the matrix, as described in [23,24]. The parameter values obtained from the fitting of the electrical conductivity data into the scaling law are $\phi_{c}=0.408$ vol.% and μ=1.53. The plot log(σ) vs. $\log\left(\phi -\phi_{c}\right)$ shows a straight line displaying an excellent fit with the data, while the percolation threshold is observed at 0.6 wt.% equivalent weight of CNT loading. The value of the critical exponent obtained from the fit is in good agreement with the estimated value between 1.3 and 4 from the percolation theory for a 3D conducting network in an insulating polymer matrix according to Bauhofer et al. [25]. The lower percolation threshold observed for CNT-PC-based composites can be attributed due to the much higher aspect ratio of CNT, as estimated by the Mamunya model presented hereafter [26].

#### 3.1.2. Numerical Method to Predict the Electrical Conductivity

Power-law percolation theory is generally used to predict the electrical conductivity above the critical concentration volume of the reinforcement, as illustrated by Equation (3). A numerical method is considered as an extensive reference, offering hundreds of useful and important algorithms that can be implemented into MATLAB for a graphical interpretation of conductivity at the studied range of CNT volume loading [27]. For our present study, piecewise cubic hermit interpolating polynomial (PCHIP) [28] has been used in order to predict the electrical conductivity, showing a perfect agreement with experimental values. PCHIP is a polynomial of three degrees in the interval comprising (0.17–0.31 vol.% CNT), as shown by Equation (4) and plotted in Figure 2, where $x_{1}$, a, b, c and d are parameters fixed by the PCHIP algorithm. The degree of the PCHIP polynomial may be referred to the creation of a three-dimensional system. PCHIP graph is superposed perfectly on the AC conductivity curve with a fitting error of *R*^2^ = 1; from this result, we can present PCHIP as an effective way to predict the electrical conductivity at the whole range of CNTs loading.
(4)$$f\left(x\right)=a{\left(x-x_{1}\right)}^{3}+b{\left(x-x_{1}\right)}^{2}+c\left(x-x_{1}\right)+d$$

### 3.2. Applicability and Validation of Models to Predict the Electrical Conductivity of PC-CNT Composites

#### 3.2.1. Voet Model

According to the Voet model, the conductivity of the composite relies on the possibility of non-ohmic electrical conduction in the polymer-filler conducting system [29]. The Voet model is given as follow:(5a)$$\log(\sigma_{c})=k\cdot \sqrt[{3}]{\phi }$$
where k is a constant, ϕ  is the volume fraction of the filler and $\sigma_{c}$ is the conductivity of the composite.

For the Voet model, the plot of $\log\left(\sigma_{c}\right)$ versus $\sqrt[{3}]{\phi }$ should be a straight line which is a hypothesis of electron emission where the electrons can jump from one conductive site to another one, favored by the inner space between the nanoparticles in the insulating polymer matrix [30]. From Figure 3a, it is seen that at a low volume fraction (0.17–0.34 vol.% CNTs), there is a slight flexion, leading to a plateau in the conductivity of composites which is the sign of conversion from non-ohmic to conductive ohmic conduction at a very low volume loading of CNTs. This increment in electrical conductivity is suddenly raised for $\sqrt[{3}]{\phi }\;=0.15$, confirming the formation of a percolative three-dimensional network; this increase is more pronounced in the range $\sqrt[{3}]{\phi }=0.15-0.25$. Extra addition of CNT filler has shown low to marginal increase in the conductivity of PC-CNT composite [31]. This model is not favorable to be applied in our study, because our composite does not follow electron diffusion theory, relying on electron emission in the studied range of CNTs loading due to the interconnected CNTs pathways; when taking into account the first data point where the composite is not reinforced by CNT (pristine PC, 0.0 vol.%), a straight line cannot be obtained.

PC-CNTs composites showed a transition from non ohmic to ohmic behavior above 0.17 vol.% CNTs ensured by the conductive CNTs network in PC matrix, limiting the applicability of the Voet model. There is a remarkable linearity, as shown in Figure 3b, favored by the gradual increase in AC electrical conductivity, especially above 2.5 vol.% CNTs. In spite of electron emission in the studied range of CNTs loading conferred by the conductive CNTs pathway, the Voet model is, in our case, favorable to be applied in predicting the AC electrical conductivity of PC-CNTs composites. The Voet model can be written as:(5b)$$\log(\sigma_{c})=k\cdot \sqrt[{3}]{\phi }+\frac{\log\left(\sigma_{p}\right)}{10}$$
where $\sigma_{p}$ is the conductivity of polymer matrix in S/m.

#### 3.2.2. Bueche Model

The Bueche model is based on volume fraction and electrical conductivity of polymer and fillers. It is assumed that insulating polymer matrix acts as sol and conducting filler as a gel. According to Bueche, the electrical conductivity of a polymer-filler composite is the addition for filler and polymer of product of the electrical conductivity and volume fraction [32]. The equation based on this model is given as
(6)$$\sigma_{c}=\sigma_{f}\cdot \phi_{f}\;+\;\left(1-\phi \right)\sigma_{p}$$
where $\sigma_{p}$ is the conductivity of polymer matrix, $\sigma_{f}$ is the conductivity of filler particle, $\sigma_{c}$ is the conductivity of the composite system and $\phi_{f}$ is the volume fraction of filler.

Figure 4 exhibits the experimental and theoretical conductivity of PC-CNT composites according to the Bueche theory. From these plots, it is observed that there is a large difference between experimental conductivity and the theoretical ones, which can be explained by the huge difference between the electrical conductivity of CNTs and the insulating PC matrix. The intrinsic conductivity of the CNTs is estimated in the range of 10^4^–10^7^ S/m, and the conductivity of pure PC is 10^−12^ S/m. This model is highly recommended to be applied in composites systems constituted of components that do not differ largely in physical properties, which makes the additive mixing rule theory applicable in this case [7]. For instance, we applied a series of hyperparameter grid-searches in order to optimize the convenient conductivity of CNTs, which satisfies this model; from a series of optimization, its value is estimated to be 2368 S/m. The disruption of conductivity of CNTs can be explained by the tendency of CNTs to break and drastically increase the number of defects during the melt blending process supplied by strong mechanical stresses at hot temperature. The shear stress imparting on the surface of a CNT can induce a pulling effect (a tensile force) on the nanotube inducing fracture and surface damages of CNTs. It results in a decrease of CNTs conductivity compared to pristine MWCNT [33]. From TEM images of PC-CNT, it is observed that below the percolation threshold, CNTs are randomly dispersed in the PC matrix. As we increase the weight concentration of CNTs above the percolative network formation, CNTs make electrical connections between themselves. In addition, we observe that CNTs are aligned in the biaxial flow direction because of the hot pressing [34]. Figure 5 shows the distribution of 1 and 2 wt.% of CNTs in PC from TEM observation.

Based on TEM observation, we make the assumption that CNTs are not shortened, saving thus their physical properties, as well as their electrical conductivity, as a result limiting the applicability of Bueche model in modeling PC-CNT composites. Furthermore, the weak DC conductivity of the considered composites (details in ref. [15]) supports the assumption, since it reports no electrical contact. In case of a degradation of CNTs armature by the shear forces during the melt blending process, the Bueche model can be applied, as shown from Figure 5, conferring to CNTs lower electrical conductivity. reducing by that the great mismatch between the polymer conductivity and filler conductivity, consequently favoring the application of mixing rule theory. Assuming that the CNTs kept their nanostructured armature, a nanoparticle size characterization is considered as a speculative way that should be performed in order to approve the experimental DC conductivity results.

#### 3.2.3. McCullough Model

The McCullough model was based on the use of transport phenomena for predicting conductivity by adding modified components to the Bueche equation [35]. The equation for predicting conductivity of polymer composite according to the McCullough model is as follows:(7)$$\sigma_{c}=\sigma_{p}\varphi_{p}+\sigma_{f}\varphi_{f}-\left[\frac{\lambda \phi_{p}\phi {\left(\sigma_{f}-\sigma_{p}\right)}^{2}}{N_{f}\sigma_{f}+N_{p}\sigma_{p}}\right]$$
where $\sigma_{c}$ is the conductivity of the composite, $\sigma_{f}$ is the conductivity of the filler, $\sigma_{p}$ is the conductivity of pure PC, $\phi_{f}$ and $\phi_{p}$ are volume fraction of the filler and polymer respectively. $N_{f}$ and $N_{p}$ are defined as:(8)$$N_{f}\;=\;\left(1-\lambda \right)\phi_{f}\;+\;\phi_{p}\lambda$$
(9)$$N_{p}\;=\;\left(1-\lambda \right)\phi_{p}\;+\;\phi_{f}\lambda$$
where λ is a structure factor that indicates the extent of conducting chain and network formation; its value varies from 0 to 1 [36]. Theoretical conductivity of the composites obtained from the McCullough model along with experimental ones is depicted in Figure 6.

The theoretical conductivity largely depends on the λ value, which is again dependent on the filler shape, size, aspect ratio and its concentration in the composite. Therefore, the exact value of λ is very difficult to calculate [37]. When λ=0 and λ=1, the theoretical conductivity is higher than experimental ones. The value of λ is optimized using a series of hyperparameter optimization; the optimized value is λ=0.91. This value closely resembles experimental data at high content of CNTs; the agreement among theoretical and experimental results is observed beyond the percolation threshold. This quantification of high CNTs chain branching above the percolation concentration is proven by TEM images [34]. An example of the entanglement of CNTs inside the polymer matrix is shown in Figure 5. By the mean of hyperparameter optimization, we have successfully simulated the degree of CNTs chain branching; this value λ=0.91 is evidence of a creation of a higher linkage between CNTs nanoparticles in PC matrix.

Figure 6 shows the plots of $\log\left(\sigma_{\mathrm{AC}}\right)$ vs. ϕ for different values of λ compared with the experimental values. The theoretical curve fits with measurement data for λ=0.91. Then, this filler shape factor λ=0.91 and as we increase the filler volume concentration beyond percolation threshold at $\phi_{c}>0.408$ vol.% of CNTs, the applicability of the McCullough model to predict PC-CNT conductive composites is favored as the value of fitting the error for the McCullough model is $R^{2}=0.9771$ and this is verified above percolation assisted by the tendency of CNTs to create long-chain linkage above the critical volume of the percolation threshold.

#### 3.2.4. Mamunya Model

Mamunya et al. [38] have studied the conductivity of composites beyond the percolation threshold based on the filler concentration in different polymers to evaluate the influence of polymer–filler interactions on the conductivity as it is represented in Equation (10).
(10)$$\log\sigma_{c}=\log\sigma_{\mathrm{phi},c}+\left(\log\sigma_{\max}-\log\sigma_{\mathrm{phi},c}\right){\left(\frac{\phi -\phi_{c}}{F-\phi_{c}}\right)}^{k}$$
where $\sigma_{c}$, $\sigma_{\mathrm{phi},c}$ and $\sigma_{\max}$ are the electrical conductivity of the composite, of the composite at the percolation threshold, of the composite at a maximum volume fraction of filler, respectively. F is attributed to the maximum volume fraction of CNTs loading used in our study and will be defined below using Equation (10). Meanwhile, $\sigma_{\mathrm{phi},c}$ is determined precisely using the PCHIP numerical method, as described in Section 3.1.2; its value is estimated to $\sigma_{\mathrm{phi},c}=3.71\;S/m$. The use of the Mamunya model to predict the experimental values of the conductivity has been firstly investigated. Figure 7 shows that the Mamunya model is able to predict the electrical conductivity of PC-CNT samples above the critical percolation volume fraction by a fitting error $R^{2}=0.92$. This reinforces our results obtained from classical percolation theory discussed in Section 3.1.1, as a Mamunya model is well fitted above $\phi_{c}=0.408$ vol.%, resulting in k=0.7; its mathematical model can be written as follows:(11)$$\log\sigma_{c}=\log\left(3.71\right)+\left(\log\left(90\right)-\log\left(3.71\right)\right){\left(\frac{\phi -0.00408}{0.031-0.00408}\right)}^{0.70}$$

High electrical conductivity is attributed to the high CNTs conductive network formation, as assessed by McCullough model and proven by TEM images. It can also be attributed to the high aspect ratio of the CNTs nanofiller; CNTs can be shortened by lowering the respective value of the aspect ratio using high shear rate of mixing—as a result, high loading of CNTs is required to have a percolating network [39]. The maximum volume fraction of the filler (F) presented in Equation (12) can also be expressed by function of geometric aspect ratio of the filler as [40]:(12)$$F=\frac{5}{\frac{75}{10+\mathrm{AR}}+\mathrm{AR}}$$

In order to simulate the performance of our preparation of the composite, we have focused on circumscribing the range of aspect ratio of MWCNTs in PC composites, since its maximal value in the PC matrix is not reported. The aspect ratio into the Mamunya model using Equation (12) and the grid-search hyperparameter optimization of multiple variable (k, AR and $\phi_{c}$) is integrated. A series of optimizations varying the aspect ratio from 150≤AR≤200 in order to get the optimal aspect ratio of carbon nanotubes in PC-based conductive composites was made. The same values of k and $\phi_{c}$ as shown in Equation (11) have been obtained; reinforcing by that Mamunya fundamental model (Equation (10)) and classical percolation theory for predicting percolation threshold volume concentration of CNTs. The same value of exponent factor k can be attributed to the effective predictive way of monitoring precisely the percolation threshold $\phi_{c}$ using grid search optimization of the implemented Mamunya algorithm model, which is in function of AR in this case; and to the maximal volume fraction of the filler that is not exceeding 3.1%.vol for the calculated range of aspect ratios.

It is observed from Figure 8 that the experimental electrical conductivity follows scaling the Mamunya model lines at aspect ratio in the range 150≤AR≤200; the superposed experimental conductivity values are more pronounced for an optimized design AR=167 obtained from hyperparameter optimization, and this agreement is more obvious beyond percolation threshold. This value is in agreement with the Gaussian distribution value of aspect ratio (AR≥150) delivered by the fabricant of multi-walled carbon nanotubes (Nanocyl in Belgium [15]). The simulated optimized value of aspect ratio is 167; at this value, a good fit between theoretical and experimental electrical conductivity data is observed. This higher value of aspect ratio may be attributed to the low critical volume of CNT for the formation of percolating pathway [41,42] and to the fine dispersion of CNTs in the PC matrix, as proven by the improvement of electrical conductivity [43].

Meanwhile, for an optimized value of aspect ratio maximum volume, fraction F of the CNT is equivalent to the weight CNTs fraction used to reach optimal electromagnetic absorption index [14], as also depicted in Section 3.3. From Figure 8, we can conclude that the way of preparation of PC/CNT samples is efficient according to the perfect match of the theoretical electrical conductivity at an aspect ratio ranging from 150≤AR≤167, proving that our operational extrusion conditions have not altered CNTs chain network. For the optimized aspect ratio the value of the fitting error is *R*^2^ = 0.967 implying the feasibility of our presented model as presented in Equation (13) to predict the AC electrical conductivity.
(13)$$\log\sigma_{c}=\log\left(3.71\right)+\left(\log\left(90\right)-\log\left(3.71\right)\right){\left(\frac{\phi -0.00408}{0.029-0.00408}\right)}^{0.70}$$

The exponent factor k of Mamunya model in Equation (10) can also be expressed as a function of a surface energy of the polymer and the nanofiller, as it can be expressed by Equations (14) and (15), where $\gamma_{p}$ and $\gamma_{f}$ are surface energies of polymer and fillers respectively; A and B are two fitting parameters [40,44]. The surface energy of PC and pristine multi-walled carbon nanotubes is 45 and 42.2 mJ/m^2^ respectively; it is determined using a contact angle goniometer (model: 100-00-(115/220)-S, Rame–Hart–German [45]).
(14)$$k=\left(A-B\cdot \gamma_{\mathrm{pf}}\right)\times \frac{\phi_{c}}{{\left(\phi -\phi_{c}\right)}^{0.75}}$$
(15)$$\gamma_{\mathrm{pf}}=\gamma_{p}+\gamma_{f}-2{\left(\gamma_{p}\gamma_{f}\right)}^{0.5}$$

Our approach is based on the integration of Equations (12), (14) and (15) in the fundamental model of Mamunya as a predictive tool for the electrical conductivity. The goal of our strategy is to model the exponent factor k as a function of the surface energy of the composite, the filling charge and the critical volume fraction for percolation; as its value is reported not to be a universal value and can be adjusted in order to have the same superposition, as Equation (13) shows in [11]. By means of the multiple grid-search hyperparameters optimization, the value of MSE is minimized and the value of the squared fit error *R*^2^ is maximized, thus approaching the same fit error, obtained from Equation (14), as shown in a comparative performance of the coefficient of fitting *R*^2^ between the two Mamunya algorithms from Figure 9a,b presenting k as an independent exponent factor on one hand (Equation (10)), and in another hand, as a parameter function of various factors, as shown from the integration of Equations (14) and (15) into the Mamunya model. The optimal values of A and B obtained from a series of an hyperparameter optimizations are found to be 0.11 and 0.03 respectively, as shown in Figure 10 depicted by the black mark, resulting in low mean square error value (MSE=0), hence approaching at this optimized values the experimental electrical conductivity; at these values of A and B, it is resulted a linear behavior of electrical conductivity as obtained in Figure 8 approaching by this result the same coefficient of determination *R*^2^ as obtained from Equation (13). A and B are generally defined for composites with different polymer matrices charged by a unique nanofiller; in our case, by using a hyperparameter optimization and MSE method for a composite formed by a single polymer matrix. These values do not largely differ from the ones obtained from an experimental approach [46]. The respective values of exponent’s k factors are then recorded in Figure 11. For an optimized design, an extensive Mamunya model can be written as:(16)$$\log\sigma_{c}=\log\left(3.71\right)+\left(\log\left(90\right)-\log\left(3.71\right)\right){\left(\frac{\phi -0.00408}{0.029-0.00408}\right)}^{0.109\times \frac{0.00408}{{\left(\varphi -0.00408\right)}^{0.75}}\;}$$

It shows in Figure 12 that there is a good fit between theoretical and experimental electrical conductivity at AR = 167, as assessed by $R^{2}=0.91$. As long as the value of AR is raised, the predictive optimization process is unlikely to succeed, confirming the strong dependency of Mamunya model in our study on two factors: percolation threshold and aspect ratio of the filler. The decreasing value of the fitting error may be also attributed to the value of composite surface energy, since its precise value is not reported by Nanocyl Company.

In this work, the Mamunya model has been thoroughly discussed and well developed. Combining some basic information of the composites, e.g., the AR of the fillers and the surface energy between the fillers and the polymer matrix, the conductivity of the composite is able to be precisely predicted using the developed Mamunya model, which provides some guidance of the design of the composites in the future.


#### 3.2.5. Sohi Model Extension

Sohi is a statistical mathematical model derived from the Scarisbrick model by the introduction of the filler aspect ratio (A), surface to volume ratio in µm^−1^ (S) and the conductivity of the polymer matrix [36], as follows:(17)$$\sigma_{c}=C\cdot \sigma_{f}\frac{A}{10}S\cdot \phi \cdot {\left(\phi \right)}^{\phi^{\frac{-2}{3}}}+\left(1-\phi \right)\sigma_{p}$$
where C is a geometric factor describing the geometry of the filler in the polymer matrix; its values are between 1 and 0.001.

Theoretical conductivities according to Sohi model are calculated for an aspect ratio of 150 and surface to volume ratio of 427 µm^−1^ (from a data sheet provided by NanoCyl). Figure 13 displays the experimental and the theoretical conductivity of PC-CNT simulated for a geometric factor varying from 1 to 0.001 according to the Sohi model. From these plots, it can be concluded that the Sohi model fails to predict the electrical conductivity of the PC-CNT composites that can be explained by the mathematical nature of this model that is essentially governed by the conductivity of the polymer matrix, as seen by the plots in Figure 13. In the present work, we are using a low volume concentration of carbon nanotubes; the first term of Equation (17) has no significant improvement in the conductivity of the composite.

Our team has proposed to maximize the aspect ratio and to minimize the geometric factor based on our previous results and TEM images. The same rearrangement is done regarding the filler loading, in order to maximize the first representative term ascribed to the physical characteristics of filler. As a result, the modified Sohi model can be written as follows:(18)$$\sigma_{c}=C^{2}\cdot \sigma_{f}\cdot A\cdot S\cdot \phi^{\frac{4}{3}}+\left(1-\phi \right)\sigma_{p}$$

The model presented by Equation (18) is able to predict the AC electrical conductivity with a fitting error equal to $R^{2}=0.922$. Additionally, the effect of varying the surface to volume ratio has been studied. As shown in Figure 14, it can be concluded that our proposed model based on hyperparameter optimization using MSE method is highly verified at a surface to volume ratio $S=428\;{\mu m}^{-1}$, since carbon nanotubes in PC matrix have maintained their surface to volume ratio supporting the feasibility of our melt blending procedure. For the simplest viewpoint, the optimized value of the geometric factor is obtained from our extensive algorithm, as expressed in Equation (18); its value C is ranging from 0.001 to 1 as depicted in Figure 15. Sohi model extension may be written as follows:(19)$$\sigma_{c}=C^{2}\cdot \sigma_{f}\cdot \frac{A}{10}\cdot S\cdot \phi^{\frac{4}{3}}+\left(1-\phi \right)\sigma_{p}$$

The aim is to use Equation (19) as a standard model issued essentially from a mathematical, structural and hyperparameter optimization approach. In further research related to the prediction of electrical conductivity of composites. Our model, as evaluated by Equation (19), may be simply recommended and applied.

### 3.3. Electromagnetic Absorption Performance

CNT–PC-based conductive composites are the basis for the wideband electromagnetic absorbers developed in this work. The experimental characterization of PC-CNT monolayer systems is performed by a waveguide measurement, as described in the test and characterization part. Electromagnetic absorption index is calculated from measured S–parameters according to [47]:(20)$$A=1-{\left|S_{21}\right|}^{2}+{\left|S_{11}\right|}^{2}$$

The Steffen–Boltzmann (SB) is an exponential regression model that can be used for such composite systems to determine the percolation threshold for different electrical properties like conductivity, dielectric constant, and absorption A, as expressed in the following equation [13]
(21)$$Y=A_{2}+\frac{A_{1}-A_{2}}{1+exp^{\left(\frac{x-x_{0}}{\Delta x}\right)}}$$

Hereafter, we will investigate the use of SB to model the electromagnetic absorption index and predict the percolation threshold. SB equation can be written as:(22)$$A\left(\%\right)=A_{2}+\frac{A_{1}-A_{2}}{1+exp^{\left(\frac{\phi -\phi_{c}}{\Delta \varphi }\right)}}$$
where $A_{1}$, $A_{2}$ are the initial and final electromagnetic absorption indices; φ and $\varphi_{c}$ are the volume fraction of the filler and at the percolation threshold respectively, and Δφ is defined as constant fitting. The results of the fitting curves plotted in Figure 16 are summarized in Table 2.

Table 2 shows the different Steffen–Boltzmann parameters obtained from fitting the electromagnetic absorption index against volume fraction at 15 and 25 GHz. The electromagnetic absorption ratio of PC-CNT composites shows a perfect fitting at 25 GHz, as the value of squared error is a unity and the error associated with other parameters (shown as second line in Table 2) is close to zero [18]. Unlike A_1_, which refers to the initial electromagnetic absorption index of the insulating polymer matrix which is quite difficult to predict by the SB model due to its lowest value. The value of percolation threshold using SB model is around 0.1 vol.% of CNTs less than the critical volume fraction obtained from the power-law of electrical conductivity model; replacing this value $x_{c}=0.1\;\mathrm{vol}.\%$ in the logarithmic power law scaling line, the magnitude of exponent factor ascribed to the dimensionality of the network system is lowered by 1.3 in magnitude; this type of difference has also reported elsewhere [32,48]. From 0.1 to 2 vol.%, more closed packed conductive pathways of CNTs are formed, interacting with incident microwave signal and leading to an increase in the electromagnetic absorption ratio [13,34]. Its maximal values are 50% and 48% at 15 and 25 GHz respectively. From Figure 16, it is observed that increasing the frequency leads to a shifting in the electromagnetic absorption index towards higher values, inducing a smaller volume fraction of CNTs for the percolation threshold. PC-CNT composites show also frequency dependency and electromagnetic absorption more pronounced at 15 GHz. With the progressive increase in filler concentration above 2 vol.% volume fraction, electromagnetic absorption ratio reaches a plateau of saturation and further CNTs enhancement does not show an increase in absorption which makes concentration 2 vol.% the optimum filler content for electromagnetic absorption. Complementary analysis of larger CNT concentrations in polymer has been realized; it results that the ultimate EMI absorber is targeted in this range of loading [14].

## 4. Conclusions

The percolation theory is a useful tool to determine the best concentration in order to produce a composite having the desired conductivity for a given EM application and is used in this work. The superiority of power-law model and numerical method to predict the critical volume concentration and its average AC electrical conductivity respectively, and assess the limitation of the Voet, Bueche and Sohi models, is shown in our study. The electrical conductivity predicted by the McCullough and Mamunya model exhibits some similarity with the practical conductivity; these results have been reinforced with microstructural TEM images which showed a high extent of CNTs chain branching and the formation of closed packed conductive pathway at and beyond the percolation threshold. It was found that the validation of the Mamunya model is related to the characteristic value of the aspect ratio. Mamunya and McCullough models are fitted with the experimental values of electrical conductivity above the percolation threshold volume fraction. Based on MSE values, the McCullough model is more practical to predict the average AC conductivity of PC-CNT, due to its largest number of points predicting the electrical conductivity denoted by a fine superposition between the experimental and the data points obtained from the model.


Our developed models showed that their validation requires a combination of mathematical and TEM characterization tools as well as the optimized results of the Mamunya model. Overall, the use of theoretical models using hyperparameter grid optimization is a successful method for predicting the average AC electrical conductivity of PC-based low CNT concentration. The Steffen–Boltzmann model assessed the applicability of modeling the electromagnetic absorption index of PC-CNT composite films and determining the optimal volume concentration for microwave absorption at 15–25 GHz.


## Figures and Tables

**Figure 1 materials-14-01687-f001:**
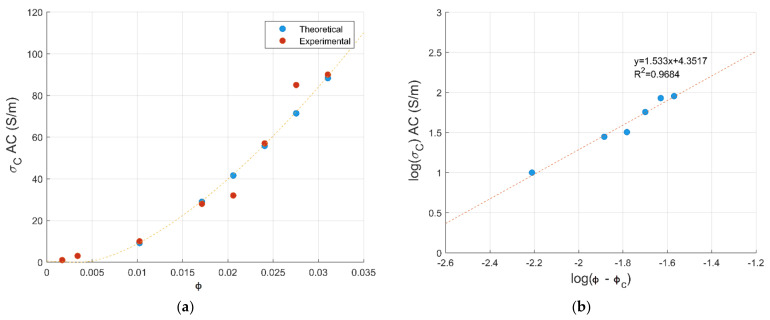
Electrical conductivity of PC-CNT vs vol.% of CNTs (**a**), and the plot of log(σ) vs. $\log\left(\phi -\phi_{c}\right)$ in (**b**).

**Figure 2 materials-14-01687-f002:**
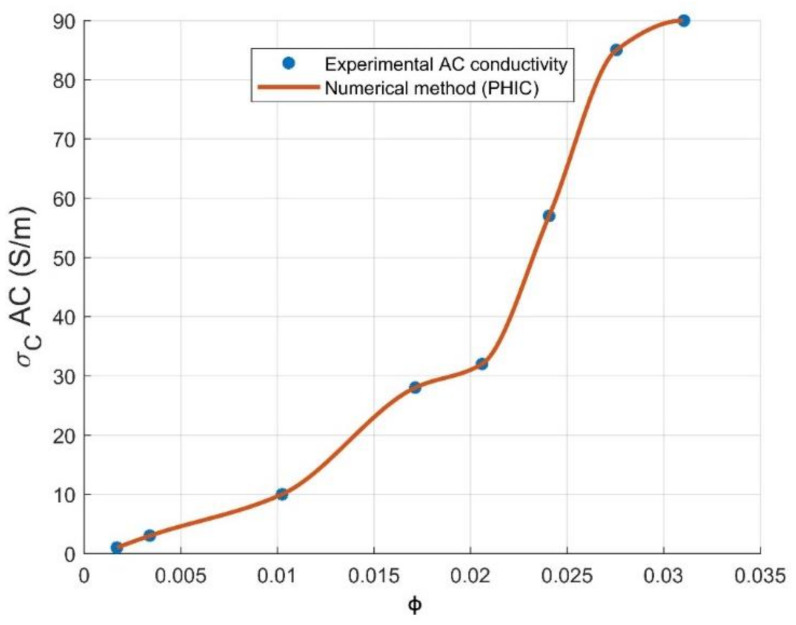
PCHIP numerical method and experimental data of electrical conductivity $\sigma_{c}$ for PC-CNT composites at several volume fraction of reinforcement ϕ.

**Figure 3 materials-14-01687-f003:**
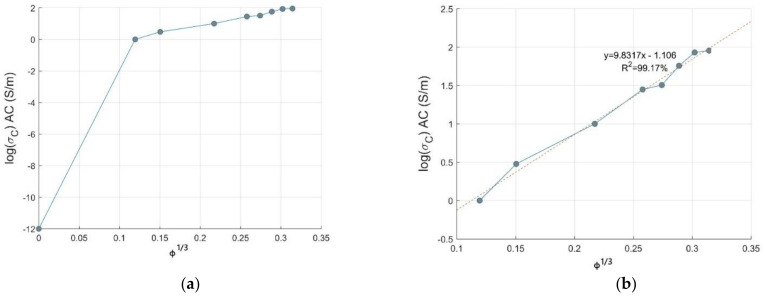
Theoretical conductivity against volume fraction of CNTs based on Voet model for PC-CNT conductive composites. (**a**) low volume fraction, (**b**) remarkable linearity.

**Figure 4 materials-14-01687-f004:**
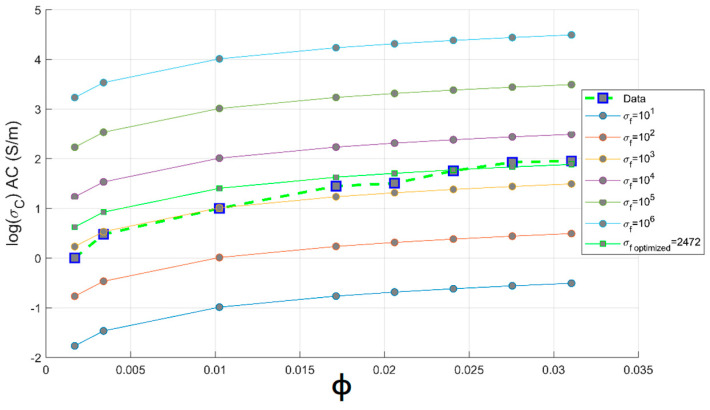
Theoretical and experimental plots of conductivity against volume fraction of CNTs based on the Bueche model for PC-CNT conductive composites.

**Figure 5 materials-14-01687-f005:**
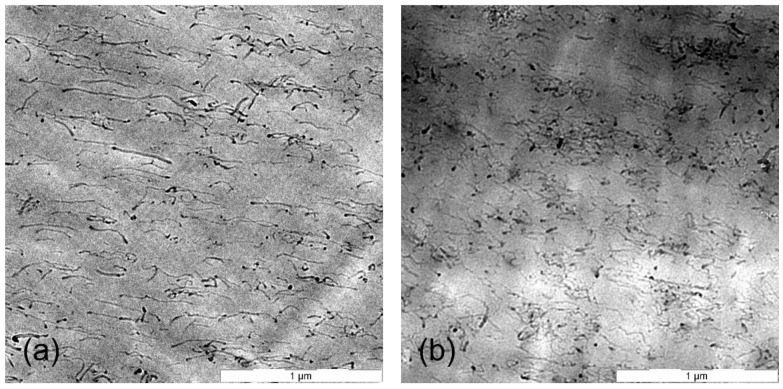
TEM observation at 8000× of a PC-1 wt.% CNT blend composite in (**a**), and a PC-2 wt.% CNT composite in (**b**).

**Figure 6 materials-14-01687-f006:**
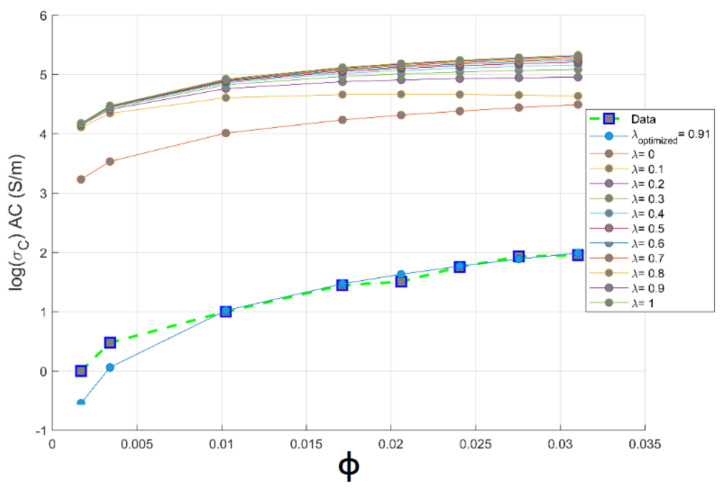
Theoretical and experimental plots of conductivity against volume fraction of CNTs based on McCullough model for PC-CNT conductive composite system.

**Figure 7 materials-14-01687-f007:**
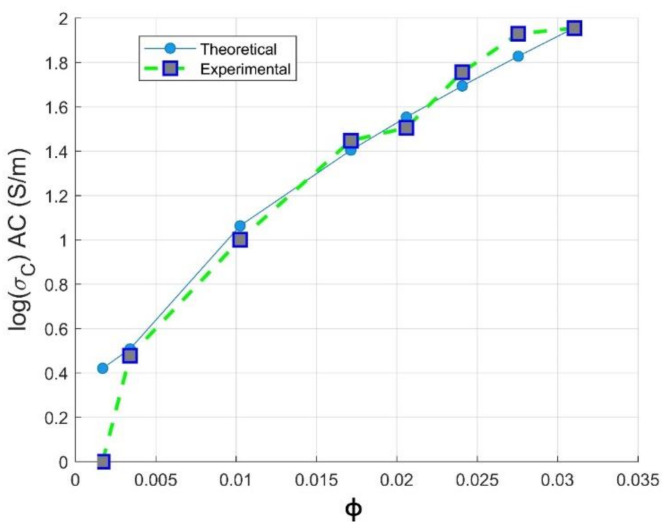
Experimental and Mamunya theoretical electrical conductivity against volume fraction of CNTs loading.

**Figure 8 materials-14-01687-f008:**
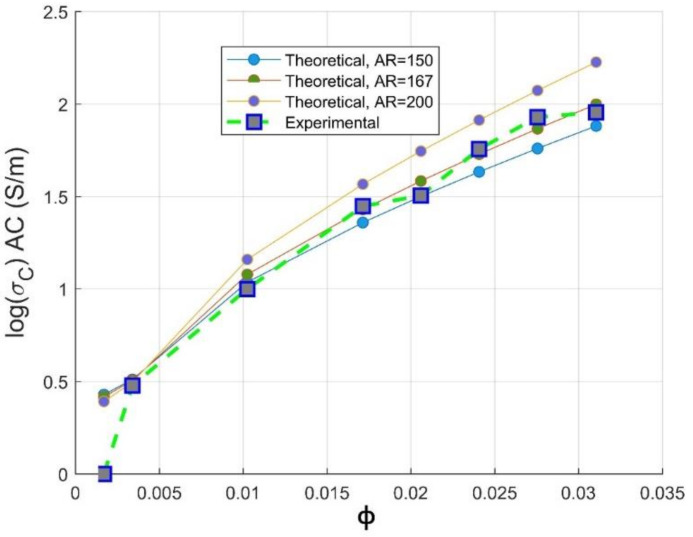
Experimental electrical conductivity and Mamunya theoretical conductivity as function of AR against CNTs volume fraction.

**Figure 9 materials-14-01687-f009:**
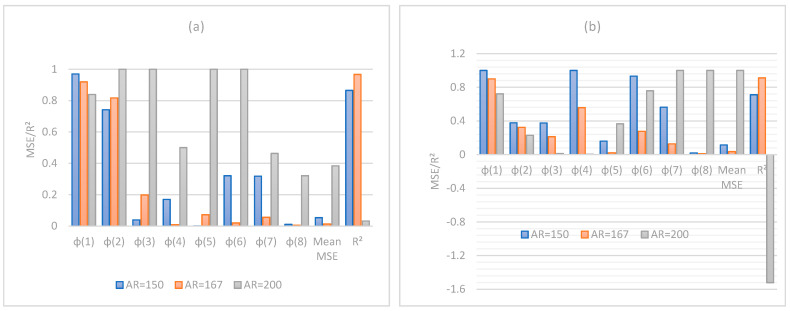
Comparative performance of MSE and $R^{2}$ for Mamunya theoretical model, as described respectively by (**a**) Equation (10) and (**b**) Equations (14) and (15). ϕ (k) values are the experimental concentrations considered in this work, thus corresponding to k samples.

**Figure 10 materials-14-01687-f010:**
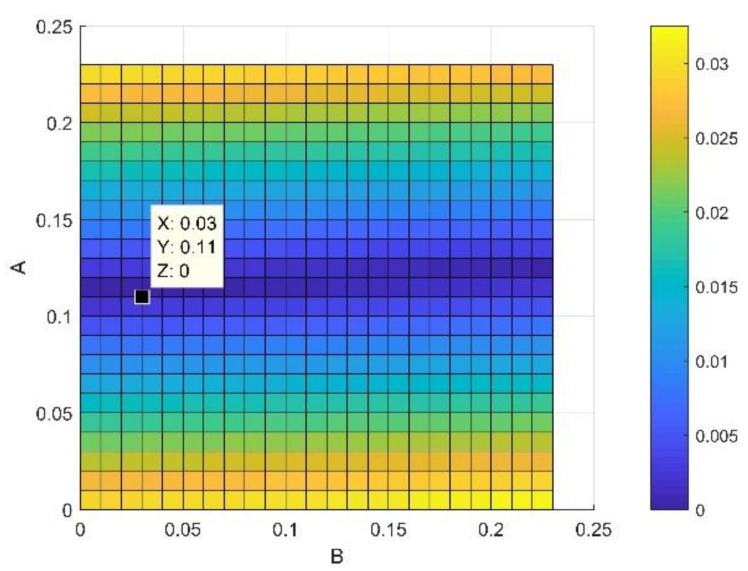
Optimization of A and B factors.

**Figure 11 materials-14-01687-f011:**
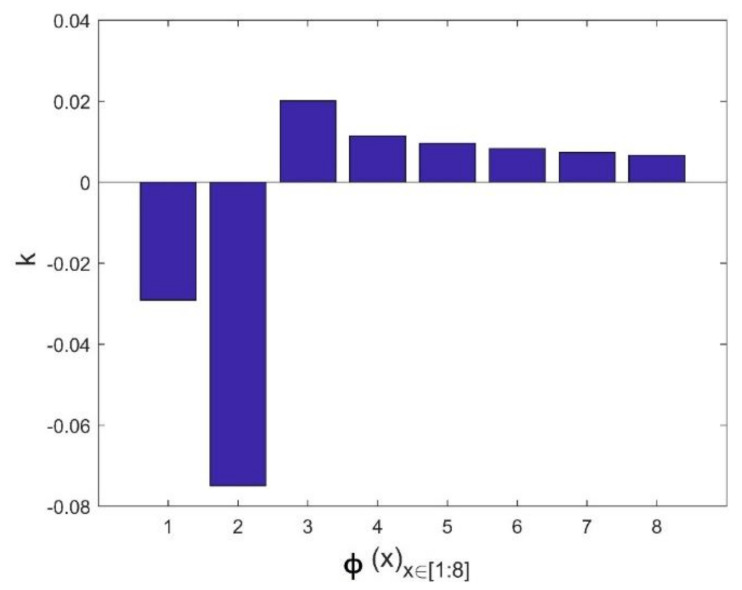
Calculated k exponent factors for minimizing the MSE factors according to optimized A and B constants.

**Figure 12 materials-14-01687-f012:**
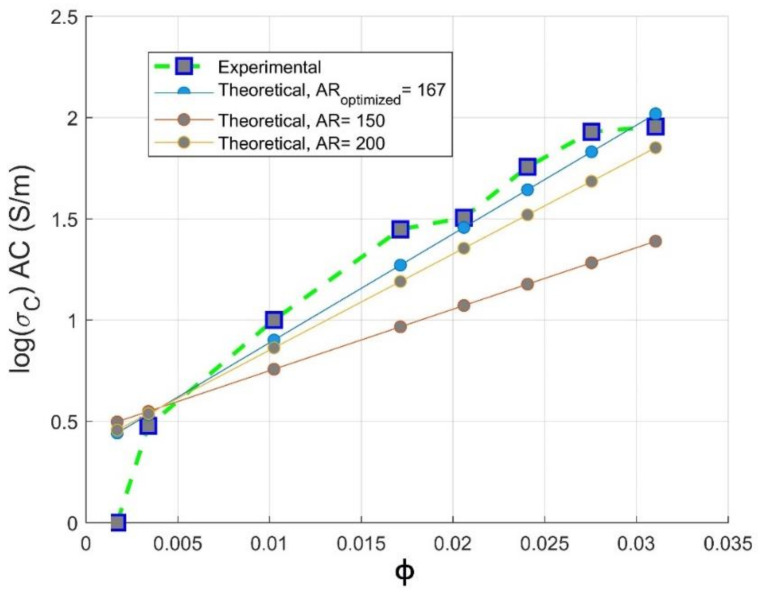
Comparative theoretical and experimental plots of conductivity against volume fraction of CNTs based on the Mamunya model for different aspect ratio for CNT nanofiller.

**Figure 13 materials-14-01687-f013:**
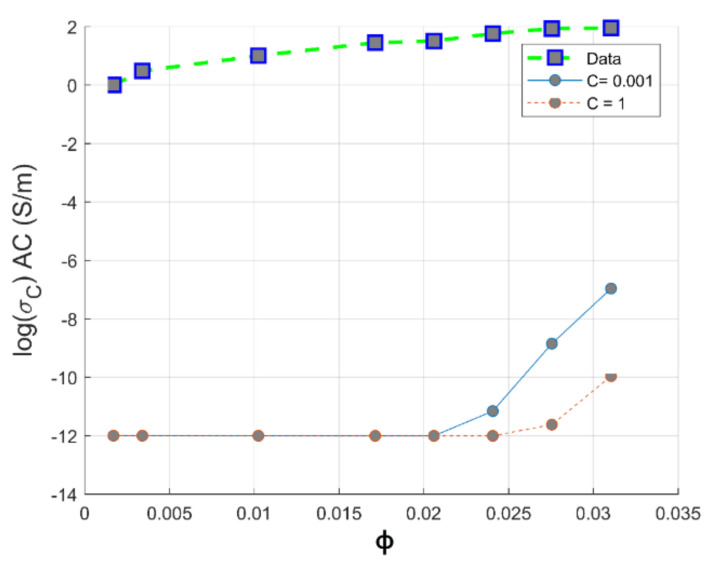
Theoretical and experimental plots of conductivity against volume fraction of CNTs based on the Sohi model for PC-CNT conductive composite system.

**Figure 14 materials-14-01687-f014:**
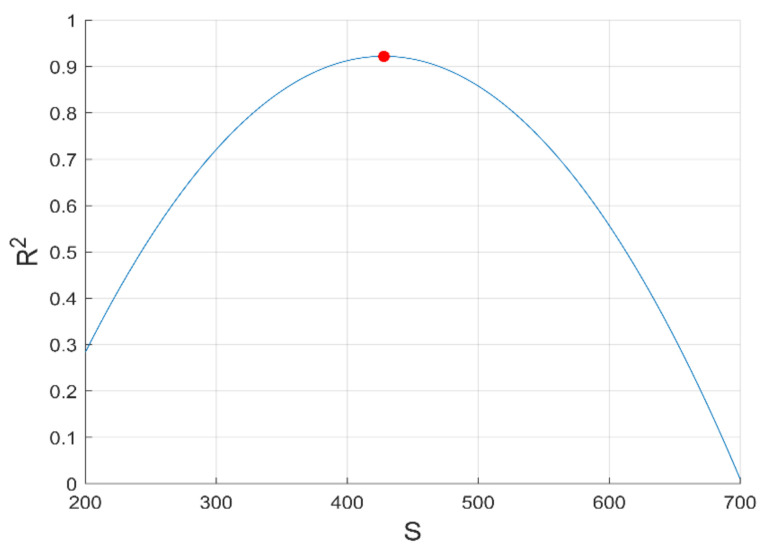
Optimization of surface-to-volume ratio of CNT nanofiller for the Sohi extension model.

**Figure 15 materials-14-01687-f015:**
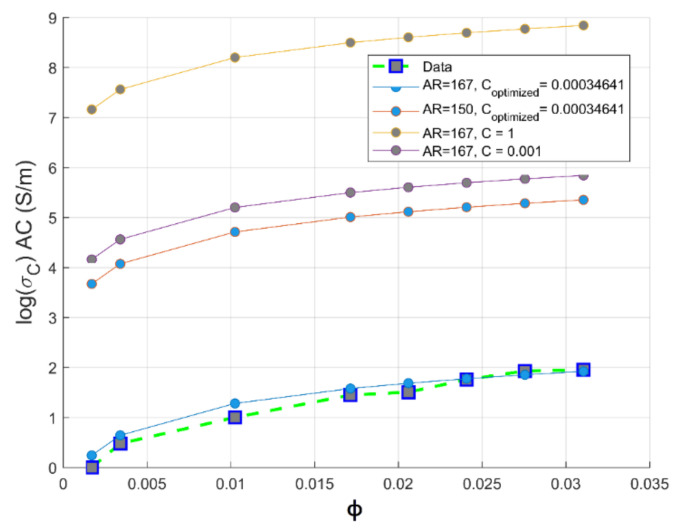
Experimental and theoretical electrical conductivity based on the modified Sohi model.

**Figure 16 materials-14-01687-f016:**
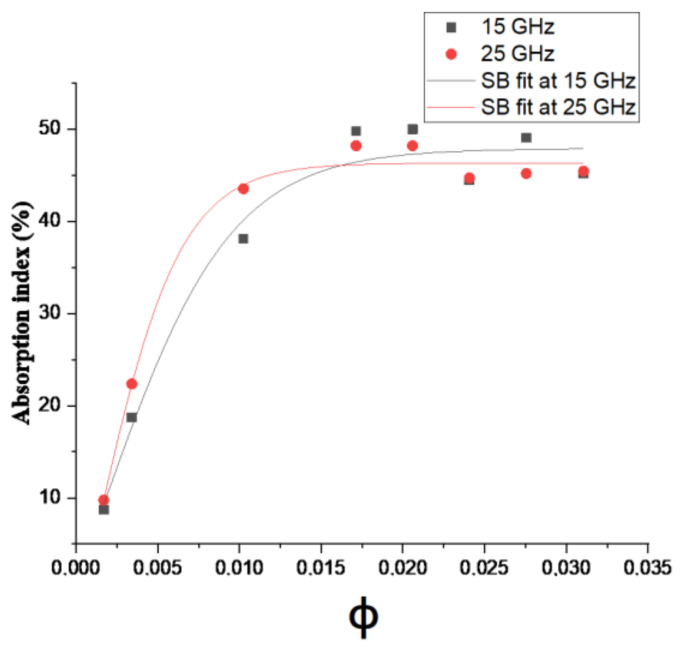
Experimental absorption ratio and Steffen–Boltzmann (SB) fit at 15–25 GHz for PC-CNT composites.

**Table 1 materials-14-01687-t001:** Concentration of CNTs in PC matrix (wt.% and vol.%).

CNTs (wt.%)	0.25	0.5	1.5	2.5	3	3.5	4	4.5
CNTs (vol.%)	0.17	0.34	1.03	1.71	2.05	2.41	2.75	3.1

**Table 2 materials-14-01687-t002:** Steffen–Boltzmann parameters values obtained from fitting electromagnetic absorption ratio against volume fraction experimental curves at 15 and 25 GHz.

Composition	Frequency [GHz]	$R^{2}$	$A_{1}$	$A_{2}$	$\varphi_{c}$	Δφ
PC-CNT	15	0.975	−30.69±107.29	47.90±1.86	0.00156±0.0102	0.00393±0.00252
	25	0.99	−37.02±77.88	46.31±0.84	0.00106±0.0045	0.0026±0.00117

## Data Availability

Not applicable.

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
