# Peer review of "Predictive Optimization of Electrical Conductivity of Polycarbonate Composites at Different Concentrations of Carbon Nanotubes: A Valorization of Conductive Nanocomposite Theoretical Models"

_materials, 2021, doi:10.3390/ma14071687_

Round 1
Reviewer 1 Report
This research study deals with carbon nanotubes as conductive fillers that introduced into Polycarbonate to form a conductive composite. However, due to the high aspect ratio of CNT its difficult to predict the effective conductivity of CNT-based nanocomposite. According to the literature, there are several theoretical models that can predict the conductivity of CNT. However. In this paper the authors focused on the use of hyperparameter grid optimization to predict the average AC conductivity, in addition, they estimate the physical properties of CNT with measured AC conductivities, and finally, they applied different theoretical models and check the validation and limitation and compared it with their experimental data which is the originality of this study and it adds value to this area of research. In general, the manuscript is well written, the text is clear and easy to read and follow. the authors address the main object of the work and the conclusions consistent with the evidence and arguments presented.
The manuscript is well written and the overall paper seems only to show some minor inconsistencies:
It is better to present Figure 10 more clearly.
The results summarized in table 2 are based on Figure 14 or Figure 17?
Author Response
We thank the reviewer for his positive evaluation and in-depth analysis. The minor inconsistences drawn by the reviewer have been taken into account and changed (see magenta color). Answers to the comments are listed below:
- It is better to present Figure 10 more clearly:
Answer: We redrew the figure 10 (renumbered fig.9) for better clarity of the legend and axis.
- The results summarized in table 2 are based on Figure 17.
Answer: The text has been modified in this sense to avoid any ambiguity. Please note that the figure has been renumbered 16.
Reviewer 2 Report
- Line 26 shows a grammar error including missing world, probably it has to be "aspect ratio".
- The sentence in lines 57-60 has the wrong grammar form.
- Names of authors given in References are totally presented wrong. Must be normalized according to Journal standards.
- Please double-check if the values of density presented in line 87 are not opposite?
- The picture presented in Fig. 6 has definitely different magnification than presented in Fig. 4b, but is shown 8000x for both?
- The paragraph from line 280 to line 287 is not clear; please use passive voice.
- Lines: 301, 320, 322, 324, 348, 352, 361, 362, 367, 379, 399, 427, 434, 435, 439, 442, 496 use passive voice. Line 302, 396 wrong grammar.
- Figure 10b the graphical image is illegible.
- Fig. 14 shows no relation between experimental data and Sohi model. There is no reference in the text.
- Do correction of Fig. 16 description; define S parameter.
Author Response
Answers to comments made by reviewer 2
First, we thank the reviewer for the comments because they have been really useful to improve the quality of the paper. Answers to the comments are listed below, while changes made are marked in green.
- Line 26 shows a grammar error including missing world, probably it has to be "aspect ratio".
Answer: done
- The sentence in lines 57-60 has the wrong grammar form:
Answer: the sentence has been modified
- Names of authors given in References are totally presented wrong. Must be normalized according to Journal standards:
Answer: All references have been checked and now meet the style of MDPI Materials journal.
- Please double-check if the values of density presented in line 87 are not opposite:
Answer: The values have been checked and we assert they are correct
- The picture presented in Fig. 6 has definitely different magnification than presented in Fig. 4b, but is shown 8000x for both:
Answer: Figures 4 have been changed based on your advice and the magnification is then identical. The text has been also complete for more information about those images. Fig.6 has been removed since judged not relevant.
- The paragraph from line 280 to line 287 is not clear; please use passive voice:
Answer: the correction has been done
- Lines: 301, 320, 322, 324, 348, 352, 361, 362, 367, 379, 399, 427, 434, 435, 439, 442, 496 use passive voice:
Answer: the correction has been done
- Line 302, 396 wrong grammar:
Answer: the sentences have been rephrased, the grammar meets better quality.
- Figure 10b the graphical image is illegible:
Answer: We redrew figure 9 (previously numbered 10) for better clarity of the legend and axis
- 14 shows no relation between experimental data and Sohi model. There is no reference in the text:
Answer: Some reference and explanations have been added in the text. We think it is increase understanding for reader.
- Do correction of Fig. 16 description; define S parameter: Please tell me what does he mean?
Answer: The S factor is the surface to volume ratio defined in equation 17. We changed the text to clarify this point.
Reviewer 3 Report
General comments
Well written manuscript. Needs minor revisions.
Specific comments
The self citation in the statement is unnecessary. "The density of PC and MWCNTs are 1.19 g/cm3 and 1.75 g/cm3 respectively [14]."
The maximum concentration of CNT used in this study is 4.5 wt%, but TEM of PC-7.5 wt.% CNT are exhibited in Fig 4(b) and 6. Either remove this TEM of PC-7.5 wt.% CNT or include conductivity data of PC-7.5 wt.% CNT in revised manuscript.
TEM of PC-7.5 wt.% CNT are exhibited in both Fig 4(b) and 6 which is rudimentary. Fig. 6 can be removed.
Authors say "Based on TEM observation, we conclude that CNTs are not shortened.." which is speculative in light of no systematic particle size analysis has been carried out in manuscript. Rephrase your speculative statement.
Replot Fig. 10(b).
Author Response
Answers to comments made by reviewer 3
We thank the reviewer for the positive evaluation and in-depth analysis. The minor inconsistences drawn by the reviewer have been taken into account and changed. Answers to the comments are listed below while changes are marked in yellow.
- The self-citation in the statement is unnecessary. "The density of PC and MWCNTs are 1.19 g/cm3 and 1.75 g/cm3 respectively [14]
Answer: Indeed, this citation has been removed because not real relevant.
- The maximum concentration of CNT used in this study is 4.5 wt%, but TEM of PC-7.5 wt.% CNT are exhibited in Fig 4(b) and 6. Either remove this TEM of PC-7.5 wt.% CNT or include conductivity data of PC-7.5 wt.% CNT in revised manuscript:
Answer: The figures have been changed to correspond to the considered composites. Now TEM images are PC-CNT2v.% and PC-CNT1v.%CNT composites.
- TEM of PC-7.5 wt.% CNT are exhibited in both Fig 4(b) and 6 which is rudimentary. Fig. 6 can be removed:
Answer: The figure 6 has been removed for this reason.
- Authors say "Based on TEM observation, we conclude that CNTs are not shortened." which is speculative in light of no systematic particle size analysis has been carried out in manuscript. Rephrase your speculative statement :
Answer: The TEM observation is presented now as an assumption. This hypothesis is comforted by the weak DC conductivity of the considered composites (details in ref.[15]).
- Replot Fig.10(b).
Answer: the whole figure 9 (previously numbered 10) has been redrawn.
Reviewer 4 Report
The manuscript by Salah et al. explores an efficient way to optimize the conductivity of polycarbonate – carbon nanotube composites with different CNT contents. The manuscript is written well. The figures have acceptable quality and are supported with sufficient discussions, although the text needs extensive grammatical and typo corrections. Therefore, I’m willing to recommend this manuscript for publication in Materials. There are some issues to be solved before the final acceptance and publication:
1- Please define acronyms and abbreviations when they first appeared in the text. For example: AC in the abstract (line 18)
2- Line 26: please correct the text: “…. and the aspect are is estimated to…..”
3- Line 32: Please delete the abbreviations in the keywords for “polycarbonate” and “carbon nanotubes”
4- Line 37: Please correct the text to “ …carbon nanotube-based composites…”
5- Line 45: Please modify the below text to: “…the electrical behavior of composite materials are…”
6- Line 47: Please add suitable references to the below statement: “Several theoretical models have been applied to predict the conductivity of the polymer–nanofiller composite systems.”
7- Line 49: Please delete “on the other hand”
8- Line 54: Please edit the refs as follows: [6,7]
Line 142: [13,21]
Line 152: [22,23]
9- Line 83: Please delete “polycarbonate”. You have already defined it as PC. Please check whole the manuscript and correct the inconsistencies accordingly.
10- Inset of Figure 1a: Please correct the inset to “Theoretical”
11- Line 93: Do you mean vol.%? It’s better use vol.% instead of v.% throughout the text.
12- Line 164: Please rewrite the below sentence. It’s unclear. “Power-law percolation theory is used mathematically to predict generally the electrical conductivity above the critical concentration volume of the reinforcement as illustrated by equation (3).”
13- Line 205: Please rewrite Figure caption 3. It’s difficult to follow the descriptions for Figures 3a and 3b.
14- Line 213: Please define sp in Equation 5b.
15- Lines 222 and 242: The order of Figures 4 and 5 should be changed, as Figure 5 is discussed in the text sooner than Figure 4.
16- Figure 5, line 250. The quality of image is not acceptable. Larger fonts should be used to be eligible for the readers. The thickness of curves should also be increased.
17- Figure 4: The current TEM images don’t support properly the discussions in lines 237-242. Please provide stronger evidence and improve the quality of TEM micrograph to fully justify the claims mentioned in this paragraph (Page 7).
18- Lines 243 to 249: This paragraph is unclear. Please rewrite the text again.
19- Again, Figures 6 and 7 didn’t mention in the text correctly. Line 263 remarks Figure 7, but Figure 6 is discussed in line 273. Please correct the order of Figures.
20- What are the differences of Figures 4b and 6?
21- Line 299: Please explain Equation 11 before Equation 12.
22- Line 317: Please define the parameters in Equation 12.
Author Response
Answers to comments made by reviewer 4
The manuscript by Salah et al. explores an efficient way to optimize the conductivity of polycarbonate – carbon nanotube composites with different CNT contents. The manuscript is written well. The figures have acceptable quality and are supported with sufficient discussions, although the text needs extensive grammatical and typo corrections. Therefore, I’m willing to recommend this manuscript for publication in Materials. There are some issues to be solved before the final acceptance and publication:
We thank the reviewer for the positive evaluation and in-depth analysis. The minor inconsistences drawn by the reviewer have been taken into account and changed. Answers to the comments are listed below while changes are marked in blue.
- Please define acronyms and abbreviations when they first appeared in the text. For example: AC in the abstract (line 18):
Answer: AC is defined as alternative current
- Line 26: please correct the text: “…. and the aspect are is estimated to…..”:
Answer: The correction is done.
- Line 32: Please delete the abbreviations in the keywords for “polycarbonate” and “carbon nanotubes” :
Answer: The abbreviations are given from the abstract and introduction. Now, AC, DC, PC, CNT and CPC are clearly defined as alternative current, direct current, polycarbonate, carbon nanotubes and conductive polymer composites respectively.
- Line 37: Please correct the text to “…carbon nanotube-based composites…”:
Answer: The correction is done.
- Line 45: Please modify the below text to: “…the electrical behavior of composite materials are…”:
Answer: The correction is done.
- Line 47: Please add suitable references to the below statement: “Several theoretical models have been applied to predict the conductivity of the polymer–nanofiller composite systems.”
Answer: Two references have been added
- Line 49: Please delete “on the other hand”
Answer: The correction is done.
- Line 54: Please edit the refs as follows: [6,7] :
Line 142: [13,21] :
Line 152: [22,23]
Answer: all corrections are done.
- Line 83: Please delete “polycarbonate”. You have already defined it as PC. Please check whole the manuscript and correct the inconsistencies accordingly.
Answer: we carefully checked the use of PC for polycarbonate in the text.
- Inset of Figure 1: Please correct the inset to “Theoretical” :
Answer: the inset of figure 1a has been corrected into ‘theoretical’
- Line 93: Do you mean vol.%? It’s better use vol.% instead of v.% throughout the text. :
Answer: v.% has been changed into vol.% throughout the manuscript
- Line 164: Please rewrite the below sentence. It’s unclear. “Power-law percolation theory is used mathematically to predict generally the electrical conductivity above the critical concentration volume of the reinforcement as illustrated by equation (3).”
Answer: the sentence has been clarified.
- Line 205: Please rewrite Figure caption 3. It’s difficult to follow the descriptions for Figures 3a and 3b.:
Answer: The captions are changed
- Line 213: Please define pin Equation 5b.
Answer: is now defined as the conductivity of polymer matrix in S/m.
- Lines 222 and 242: The order of Figures 4 and 5 should be changed, as Figure 5 is discussed in the text sooner than Figure 4:
Answer: the order of the figures has been taken into account.
Figure 5, line 250. The quality of image is not acceptable. Larger fonts should be used to be eligible for the readers. The thickness of curves should also be increased.
Answer: The global size of the figure has been increased.
- Figure 4: The current TEM images don’t support properly the discussions in lines 237-242. Please provide stronger evidence and improve the quality of TEM micrograph to fully justify the claims mentioned in this paragraph (Page 7):
Answer: The TEM images have been changed.
Answer: we did not want to reproduce the TEM images that have been published in the referenced paper by Mefsin et al. We hope that these TEM images proves the creation of a linkage between CNT chains as proved at 2wt% above the percolation threshold and this linkage are in one directional flow.
- Lines 243 to 249: This paragraph is unclear.
Answer : the text is now adapted.
- Again, Figures 6 and 7 didn’t mention in the text correctly. Line 263 remarks Figure 7, but Figure 6 is discussed in line 273. Please correct the order of Figures
Answer: the text has been adapted.
- What are the differences of Figures 4b and 6:
Answer: we have deleted fig.6 according to the request of reviewer 3
- Line 299: Please explain Equation 11 before Equation 12:
Answer: done
- Line 317: Please define the parameters in Equation 12:
Answer: we have defined F as the maximum volume fraction of the filler
Round 2
Reviewer 2 Report
Thanks for answering 1st review requests.
Reviewer 4 Report
The manuscript is acceptable from my viewpoint in the current format. Just as an important issue, the authors should provide scale bars for TEM micrographs in Figure 5 (page 8) before the final publication.
Author Response
Dear,
we thank the reviewer for his positive evaluation , the main TEM have been added with scale bars.
Best regards
